# Psychiatric Co-Morbidities and Profile of Patients with Irritable Bowel Syndrome in Northern India

**DOI:** 10.3390/brainsci14040393

**Published:** 2024-04-18

**Authors:** Ankita Saroj, Adarsh Tripathi, Sumit Rungta, Sujita Kumar Kar

**Affiliations:** 1Department of Psychiatry, King George’s Medical University, Lucknow 226003, India; ankitasaroj.18@gmail.com (A.S.); dradarshtripathi@gmail.com (A.T.); 2Department of Medical Gastroenterology, King George’s Medical University, Lucknow 226003, India; drsumitrungta79@gmail.com

**Keywords:** irritable bowel syndrome, psychiatric co-morbidities, somatic symptoms, anxiety, stress

## Abstract

Objectives: To study sociodemographic and clinical variables, including psychiatric co-morbidities, in patients with irritable bowel syndrome. Methods: A total of 158 patients attending a medical gastroenterology clinic in a tertiary care center in Northern India were screened, from whom 100 were selected for the study. Rome IV criteria were used to diagnose IBS, and the severity of symptoms was assessed by the Irritable Bowel Syndrome Symptom Severity Scale (IBS-SSS). Psychiatric co-morbidities were screened via clinical evaluation, and if present, a diagnosis was made as per DSM-5. The Depression, Anxiety, and Stress Scale-21 (DASS-21) and Somatic Symptom Scale-8 (SSS-8) were used to assess depression, anxiety, stress, and somatic symptoms. Result: The mean age of cases was 35.6 years’ old, and the majority of cases (i.e., 38.0%) were between 18 and 29 years’ old. Males comprised 62.0% of the sample and females 38.0%. Moderate IBS was present in 61.0% of the cases. Evaluation via DASS-21 revealed that 53.0% were in the moderate category of depression, 43.0% had moderate anxiety, and 36.0% had moderate stress. The somatic symptom scale revealed that 48.0% patients were in the high category. Psychiatric co-morbidities were present in 29.0% of cases. Depressive disorders were the most common psychiatric co-morbidity. Conclusions: Patients with IBS presenting to a tertiary care center in Northern India were primarily young males living in semi-urban areas who belonged to the Hindu religion, were married, and had a nuclear family. Patients with IBS commonly have associated psychiatric disorders; anxiety disorders and depression are most common.

## 1. Introduction

Irritable bowel syndrome (IBS) is a functional gastrointestinal disorder characterized by symptoms of changes in stool form, abdominal pain, diarrhea, constipation, or a combination of the above [1]. These symptoms are common in the general population, affecting people of all ages and genders [2]. Due to the lack of reliable biomarkers, Rome IV criteria, published in May 2016, are used for diagnosis [3] and serve as the current gold standard. Compared to earlier iterations, this set of criteria prioritizes the importance of nutrition, the intestinal milieu, and the influence of cross-cultural differences [4]. The Bristol stool form scale classifies abnormal bowel movements [5].

The disorder represents a significant burden on healthcare services and accounts for almost half of the patients referred to gastroenterology clinics. The prevalence and nature of IBS symptoms differ based on geographical location due to variations in bowel habits, cultural beliefs, gut microbiota, dietary habits, and psychosocial factors [6].

The onset of IBS is more prone to manifest after an infection (IBS postinfectious) or a significant psychosocial stressor, with minimal variability observed across different age groups. The prevailing theory suggests that IBS represents a dysfunction in the interplay between the brain and the gastrointestinal (GI) tract. It has been hypothesized that alterations in the gut flora cause inflammation and decreased bowel function in at least some individuals [7].

The prevalence of IBS varies by geographic region and population, as well as the diagnostic criteria used [8]. According to cross-sectional studies conducted in Europe and North America, IBS affects 10–20% of the population. The worldwide IBS prevalence was reported to be 11.2% using Manning, Rome I, Rome II, or Rome III criteria [8,9]. Across multiple Asian countries, the occurrence of irritable bowel syndrome (IBS) varies between 4% and 20%. In India specifically, the estimated prevalence of IBS is within the range of 4.0% to 7.9%, and this rate is steadily increasing [6]. A multicentric study conducted by the Indian Society of Gastroenterology (ISG), which included 2785 patients with IBS and 4500 subjects from the community, reported an IBS prevalence of 4.2%, similar to earlier findings [6].

Between 50 and 90% of patients who are diagnosed with IBS also experience concurrent psychiatric conditions, notably anxiety disorders and depression. Research indicates that patients who seek medical assistance typically present with a greater frequency and severity of symptoms and are more prone to experiencing depression and anxiety [10].

The health-related quality of life of patients with IBS is impaired. Patients with severe conditions often have a higher rate of reduced quality of life [11]. Studies have also shown that patients avoid or are unable to participate in a variety of activities, like work, leisure, and social activities, due to IBS symptoms [11,12].

The majority of research on the relationship between IBS and associated psychiatric diseases comes from studies conducted in Western contexts. Given significant sociocultural differences influencing the manifestation of these functional disorders, extrapolating findings from Western studies [13,14,15] may lack relevance. The research that has been conducted in this area so far in India is limited. Consequently, our objective was to enhance understanding of the sociodemographic and clinical correlates, including co-morbid psychiatric disorders, associated with IBS in our specific region.

## 2. Materials and Methodology

This cross-sectional study was conducted at a tertiary care center in Northern India for about one and a half years (from June 2021 to September 2022). Approval for the study was given by the institutional ethical committee with ref. code II PGTSC-IIA/P10. The subjects who visited the outpatient services of the medical gastroenterology department were selected. A total of 158 patients were screened, and 100 were enrolled in the study. All the subjects were interviewed to evaluate their sociodemographic parameters, like gender, age, marital status, employment, and education level. A trainee psychiatric resident conducted the assessments under the supervision of a consultant medical gastroenterologist and a consultant psychiatrist. All subjects provided written consent after being fully informed about the study’s objectives and procedures and agreeing to participate in the research.

Subjects who met the Rome IV criteria, were at least 18 years’ old and below 60 years’ old, without any diagnosed medical co-morbidities, including diabetes and chronic kidney disease or pre-existing gastrointestinal disorders other than functional gastrointestinal syndromes (abnormal upper and lower GI endoscopy), and were not receiving any psychotropic medications were included in the study.

Rome IV criteria, a self-reported integrated questionnaire for diagnosing all functional gastrointestinal disorders in adults, was used to diagnose cases [3]. The irritable Bowel Syndrome Symptom Severity Scale (IBS-SSS) was used to assess the severity of symptoms. The scale comprises five questions evaluating the severity and frequency of abdominal pain, dissatisfaction with bowel habits, severity of abdominal distention, and interference with quality of life over the past ten days. These questions are rated on a 100-point visual analogue scale by the subjects [16]. The subjects were screened for any psychiatric co-morbidity during the interview, and diagnosis was confirmed by DSM-5 [17]. The Depression, Anxiety, and Stress Scale-21 (DASS-21) was applied to assess depression, anxiety, and stress symptoms. The negative emotional states of depression, anxiety, and stress are measured via three self-report scales [18]. The Somatic Symptom Scale-8 (SSS-8) was used to assess somatic symptoms. It is a brief self-report questionnaire and a shortened version of the PHQ-15 questionnaire scale [19]. The Sheehan Disability Scale (SDS) was used to check functional impairment. It is a concise, 3-item self-report tool that evaluates functional impairment in work/school, social, and family life [20]. Social and occupational functioning was assessed using the Social and Occupational Functioning Assessment Scale (SOFAS), which focuses exclusively on the individual’s level of social and occupational functioning [21]. Figure 1 outlines our methodology, our enrollment criteria, and the participants eventually selected for the study.

## 3. Statistical Analysis

Data were entered into an Excel sheet designed by Microsoft Corporation (Windows version 2019), Redmond, WA, USA. The analysis of continuous data was conducted using the mean and standard deviation. Categorical variables were expressed as percentages. The relationship between categorical variables was examined using Fisher’s exact test. The Pearson correlation was used to explore the relationship between different variables. Statistical analysis was conducted using SPSS version 25 [22].

## 4. Results

A total of 100 subjects were enrolled in the study after screening 158 subjects. The most common reason for exclusion was that the subject was already receiving psychotropic medications (n = 33), followed by subjects with medical co-morbidities like diabetes/chronic kidney disease (n = 11), a history of gastrointestinal surgery (n = 7), age > 60 years (n = 5), and abnormal upper or lower gastrointestinal endoscopy (n = 2). The sociodemographic profile of the subjects is provided in Table 1. The mean age of IBS onset for our patients was 30.46 years (SD = ±9.05), and the mean duration of illness was 5.07 years (SD = ±5.89).

In the present study, 46.0% of the cases had onset of illness in the 18–29 age group, followed by 33.0% in the 30–39 age group, 18% in the 40–49 age group, and 2% of cases in the 50–59 age group. Of the cases, 40.0% had a duration of between 1 and 3 years, while 25.0% had a duration greater than six years.

As per the IBS symptom severity scale, most subjects were in the moderate category (61.0%), and the mean score was 280.20 (SD = ±57.21). Depression, anxiety, and stress symptoms were assessed using DASS-2. On assessment, 53.0% patients had moderate depressive symptoms, with a mean score of 16.16 ± 5.31; 43.0% had moderate anxiety symptoms, with a mean score of 13.80 ± 4.76; and 36.0% had moderate stress symptoms, with a mean score of 18.53 ± 6.91. Somatic symptoms were evaluated according to the Somatic Symptom Scale-8. Of the cases, 12% were in the high category score, followed by 35.0% in medium and 28.0% in low, with a mean score of 11.50 ± 3.05. Functional impairment in the sample was assessed using the Sheehan Disability Scale. As many as 30% of subjects had impairment in work/school, with a mean score of 4.73 ± 1.70; as many as 38% of subjects had impairment in social life, with a mean score of 4.87 ± 1.65; while 51% had impairment in family life/home responsibility, with a mean score of 5.61 ± 1.59. Social and occupational functioning in the sample, as per the SOFAS score, was analyzed. As many as 31% had some difficulty in social, occupational, or school functioning, while 3% had superior functioning.

The psychiatric co-morbidities present in this study are listed in Table 2. In our study, 29% of patients had psychiatric disorders. Major psychiatric disorders seen were depressive disorders and anxiety disorders. Obsessive–compulsive disorder, conversion disorder with mixed symptoms, somatic symptom disorder, and substance use disorder were other disorders found to be present. There was no significant difference between patients with IBS with and without psychiatric co-morbidities. However, there is a considerable difference in IBS severity between patients with IBS with and without psychiatric co-morbidities. Additionally, the depressive symptom severity and somatic symptoms were significantly higher in patients of IBS with psychiatric co-morbidities in comparison to those without psychiatric co-morbidities.

Pearson correlation analyzed the association between IBS severity scores and clinical variables. IBS-SSS was significantly associated and positively correlated with age of onset (r = 0.209; *p* = 0.037), DASS 21 (depression) (r = 0.545; *p* ≤ 0.001), DASS 21 (anxiety) (r = 0.212; *p* ≤ 0.001), the Somatic Symptoms Scale (r = 0.458; *p* ≤ 0.001), and SDS (r = 0.643; *p* ≤ 0.001), and negatively correlated with SOFAS (r = −0.690; *p* ≤ 0.001).

## 5. Discussion

Our study investigated sociodemographic and clinical variables along with psychiatric co-morbidities in patients with irritable bowel syndrome. Genetic, psychological, medical, and social factors combine intricately to cause IBS, and each of these elements has the potential to impact the course of the illness [23]. Thus, the exclusion criteria screened out patients with co-morbid medical illness and pre-existing gastrointestinal disorders other than functional gastrointestinal syndromes, as these might alter gastrointestinal functioning and minimize the chances of any organicity. Subjects already receiving psychotropic medications were also excluded as it might affect the severity of IBS and can affect depression, anxiety, stress, somatic symptoms and impairment, and social and occupational functioning.

The study group comprised 100 patients whose sociodemographic details are shown in Table 1. The clinical population of IBS is middle-aged, as in our study. In our study, there were nearly twice as many males as females. These findings were similar to a multicentric study conducted by Khanna et al., who reported that the mean age of the study population was around 40 years, and most of the patients were men, outnumbering women. Patients between the ages of 31 and 45 made up about 40% of the total [24].

A multicentric study by Ghosal et al. reported the mean age of the sample as 39.4 years and a male preponderance of 68% [6]. Although Western studies [13,14,15,25] have reported higher rates of IBS in women, Asian findings [6,26] have reported higher rates of IBS in males, which could be due to easy and better access to healthcare services, as well as cultural factors favoring men in a male-dominant society. The reason for the altered rates is unidentified and could be an area for future research.

In the present study, 45.0% of the cases lived in semi-urban areas, followed by 34.0% in urban areas. Community studies conducted in South Asia (India, Bangladesh, and Malaysia) indicate that IBS is more common in urban areas than in rural populations [1]. Compared to rural living, an urban lifestyle is linked to increased psychological stress as well as other causal risk factors, such as dietary factors and a sedentary lifestyle, which could explain the reason IBS is common in urban areas.

Many subjects in our study were married, and most lived in nuclear families. Although most of the subjects in the study had attained graduate education, most were unemployed and non-earning. Earlier studies conducted by Khanna et al. [24] have also reported that a significant percentage of enrolled patients (46.0%) were graduates or postgraduates.

The higher prevalence of IBS in the age group 18–29 years, in unemployed patients, and in patients who were graduates or had received higher education could be due to higher psychological stress in these patients due to uncertainty of job, financial issues, academic stressors, and issues related to marriage. The notion that the prevalence of IBS decreases with age, indicating symptom resolution over time, is challenged by longitudinal observational studies demonstrating the persistent nature of symptoms in the long term [7,24].

In the current study, IBS severity was in the moderate category in most of the subjects. This could be due to the fact that most of the time, patients sought treatment only when their symptoms were reasonably severe. Higher severity in the participants is expected, given that the study was conducted in a tertiary care center. A previous study conducted by Lackner et al. reported a high–moderate level of IBS-SSS symptom severity, which is similar to this study [27].

Using DASS-21, depression, anxiety, and stress symptoms were assessed, revealing nearly twice the rates of depression symptoms than anxiety symptoms in the IBS subjects enrolled. Alaqeel et al. reported that the DASS-21 results in their study showed most of the patients in the moderate anxiety category [28]. Similarly, in a previous study conducted in Japan by Okami et al., medical students with IBS scored higher than control patients on the Hospital Anxiety and Depression Scale (HADS) [29]. The cases enrolled in the study had higher severity of symptoms, which may lead to more significant psychological stress, impairing the gut–brain axis [30], which is hypothesized to be disrupted in patients with IBS. Psychological distress is widely recognized as a substantial contributor to IBS, with anxiety and depression frequently linked to IBS in studies conducted earlier [31].

In the present study, SSS-8 revealed a high level of somatic symptoms in subjects. One explanation could be that compared to people in developed countries, people in developing countries report more somatic symptoms [32]. In addition, more than 50% of patients with IBS report having anxiety or depression, and these individuals have more severe somatic symptoms [31]. It is crucial that patients with IBS are evaluated for other somatic symptoms apart from GI symptoms, like pain, fatigue, and shortness of breath, as they have somatic stress in different parts and should be taken into consideration. Studies in the past have also reported a close association between somatization and IBS [33].

This study evaluated functional impairment and social and occupational functioning, which revealed that nearly half of the subjects had impairment in family life/home responsibility, and most of them had some difficulty in social, occupational, or school functioning. Owing to distressing gastrointestinal symptoms along with depressive, anxiety-related, and somatic symptoms, it is very likely for the patients to experience significant impairments in several aspects of life. We encountered limited data concerning functional impairment and social and occupational functioning. However, one study documented a notable level of impairment attributable to IBS, with nearly three-fourths of the participants reporting some degree of IBS-related impairment across at least five different domains of daily life [34]. Moreover, other studies have reported significantly poorer quality of life and more absenteeism in work in patients with IBS, which corroborates with our findings of impairment in work, social life, and family responsibilities, as well as difficulty in social, occupational, or school functioning [12].

In the present study, co-morbidity was evaluated as shown in Table 2. Diagnosable psychiatric co-morbidity was present in fewer than one-third of subjects, although psychiatric symptoms were present in a significant number of subjects. There is plenty of evidence of the prevalence and severity of IBS symptoms in individuals suffering from anxiety and mood disorders [10,35].

Consistent with our study findings, depression is the most common psychiatric condition among clinical patients with IBS, followed by anxiety disorders and somatization disorders. [35]. The outcomes observed in our research might be influenced by our study sample being drawn from a tertiary care center, where more severe cases of the illness are seen. Research has shown that the risk of co-morbid psychiatric disorders is higher in cases of functional gastrointestinal disorders. Cho et al. stated that stress, worry, and sadness are often associated with IBS; in an integrated biopsychosocial approach, psychiatric co-morbidity contributes to the etiology of irritable bowel syndrome (IBS) [11].

In the present study, the correlation of IBS-SSS with clinical variables revealed a significant correlation with all the scoring scales, i.e., DASS-21, SSS-8, SDS, and SOFAS. Subjects who scored higher on severity had more depression, anxiety, and somatic symptoms, along with more impairment in day-to-day functioning. These findings have been validated by studies in the past [27,28,31]. Similarly, a study conducted by Cho et al. concluded that severe symptoms in patients were found to demonstrate a correlation between anxiety and depression and the abdominal pain or discomfort score, although the association with anxiety was statistically insignificant [11].

Our study has a few limitations. We selected our sample from the clinical population visiting the medical gastroenterology outpatient department; hence, it might not be generalizable to other populations. Further, patients were not categorized into IBS subtypes, which could have an effect on disease severity and associated psychiatric co-morbidities. No fresh investigations were conducted at baseline due to lack of feasibility, although patients were investigated in the past. Another drawback of our study was the need for an investigation into the dietary patterns of our participants, which could have influenced the manifestation of IBS symptoms.

## 6. Conclusions

In conclusion, patients with IBS presenting to tertiary care centers in Northern India were primarily young males living in semi-urban areas who belonged to the Hindu region who, were married, and had a nuclear family. Most of the patients were graduates, unemployed, or had no earnings. They have been found to have psychiatric co-morbidities. Psychiatric symptomatology and the duration of illness are strongly associated with the severity of the disease.

## Figures and Tables

**Figure 1 brainsci-14-00393-f001:**
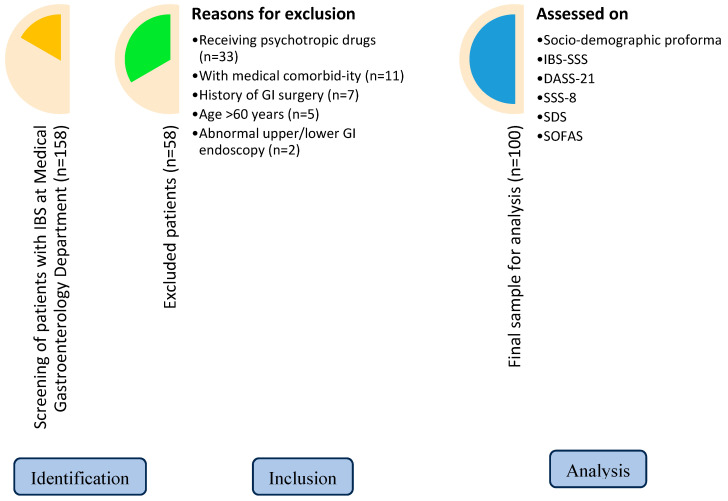
Research flowchart. GI—gastrointestinal; IBS—irritable bowel syndrome; DASS-21—Depression, Anxiety, and Stress Scale-21; SSS-8—Somatic Symptom Scale-8; SDS—Sheehan Disability Scale; SOFAS—Social and Occupational Functioning Scale.

**Table 1 brainsci-14-00393-t001:** Sociodemographic profile of patients with IBS.

Sociodemographic Variables	Number of Cases (%)
Age (in years)	
18–30	38 (38.0)
31–40	23 (23.0)
41–50	26 (26.0)
51–60	13 (13.0)
Mean (in years) ± S.D. = 35.66 ± 11.30 (range = 18–58 years)	
Gender	
Male	62 (62.0)
Female	38 (38.0)
Domicile	
Rural	21 (21.0)
Semi-urban	45 (45.0)
Urban	34 (34.0)
Religion	
Hindu	83 (83.0)
Muslim	17 (17.0)
Marital Status	
Married	71 (71.0)
Unmarried	29 (29.0)
Family type	
Nuclear	72 (72.0)
Joint	28 (28.0)
Education	
Illiterate	12 (12.0)
5th or below	7 (7.0)
6th to 8th	4 (4.0)
9th and 10th	13 (13.0)
11th and 12th	13 (13.0)
Graduate and above	51 (51.0)
Employment Status	
Unemployed *	25 (25.0)
Housewife	24 (24.0)
Unskilled/Semiskilled	16 (16.0)
Skilled	4 (4.0)
Clerk, shop owner, farmer	24 (24.0)
Professional	7 (7.0)
Patient’s income in rupees (per month)	
Nil	49 (49.0)
<10,000	7 (7.0)
10,000–25,000	13 (13.0)
>25,000	31 (31.0)

* Students are included in the unemployed category.

**Table 2 brainsci-14-00393-t002:** Psychiatric co-morbidities in patients with IBS.

Psychiatric Co-Morbidities	Number of Cases (%)
Absent	71 (71.0%)
Present *	29 (29.0%)
Depressive disorders	
• Major depressive disorder	7 (7.0%)
• Recurrent depressive disorder	4 (4.0%)
• Persistent depressive disorder	3 (3.0%)
Anxiety disorders	
• Generalized anxiety disorder	2 (2.0%)
• Unspecified anxiety disorder	7 (7.0%)
Obsessive–compulsive disorder	1 (1.0%)
Adjustment disorder	2 (2.0%)
Conversion disorder with mixed symptoms	1 (1.0%)
Somatic symptom disorder	1 (1.0%)
Substance use disorder	
• Tobacco use disorder	3 (3.0%)
• Alcohol use disorder	1 (1.0%)

* Not mutually exclusive.

## Data Availability

The data presented in this study are available on request from the corresponding author. The data are not publicly available due to ethical concerns.

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
