# Peer review of "Psychiatric Co-Morbidities and Profile of Patients with Irritable Bowel Syndrome in Northern India"

_brainsci, 2024, doi:10.3390/brainsci14040393_

Round 1

Reviewer 1 Report

Comments and Suggestions for Authors

This is a single center study in evaluating the clnical characteristics of IBS patient in North India. There are some issues that need further clarification:

1. The authors can analyze the epidemiologic data of IBS associated psychiatric disease in India and compare between the studies.

2. In patient slection, what is the definition of no medical comorbidities? Did it mean they never visit hospital and diagnosed with any diseaes before? It also include obesity? hypertension? asthma? arrhythmia? gallstone?GERD? and many common disease?

3. Did all patients receive upper and lower GI endoscopy? No patient had H.pylori infefction? If so, were they treated in recent years? H.pylori infeciton eradication may also be related with IBS, dysbiosis and psychiatric disorders.

4. What are the main treatment of IBS in India? It is commonly to use TCA, SSRI for IBS treatment, however, these drugs are excluded in this study.

5. It was strange that so many patients had psychiatric comorbidities, however, no patient were taking psychiatric medication. It hint that the clinical utility of psychiatric service was low in the hospital .

6. Who did the interviewing and diagnosis of psychiatric diagnosis? Psychiatrist or resident doctors or GI specialist?

7. Why patients age > 60 years are excluded? It wasn't shown in the article but in the figure.

8. As manly patients were illiterte or below 5th grade, how was the questionnaire completed?

9. What are the predictors of psychiatric cormobidities? The author may do univariate and multivariate analysis. 

10. Is there any difference in demographic data between patients with psychiatric disease and without psychiatric disease? 

11. How was these patients managed after entering the study, by what kind of medication or other psychiatric therapies? 

12. Was infection excluded? By what method? 

Comments on the Quality of English Language

Some typo error exist.

Reviewer 2 Report

Comments and Suggestions for Authors

I have read the manuscript titled 'Sociodemographic and Clinical Correlates Including Psychiatric Comorbidities in Clinical Population of Irritable Bowel Syndrome' with interest. The study aims to contribute to the existing literature on the characteristics of patients with IBS. The research focuses on the population of northern India, where there is a lack of data on this topic. The authors of the manuscript have clearly indicated this gap in the literature. At this point, I would like to request clarification on the term 'western studies' (line 63). It seems to refer to the region of the world from which most studies on IBS originate. Additionally, I suggest correcting the manuscript title to include an indication of the study population.

The aim of the study, as formulated in the Abstract section, raises doubts as it completely overlaps with the characteristics of the study group. To improve clarity, it is suggested that the aim of the study be modified.

Section 2 of the study's material and methodology should be supplemented with exclusion criteria. The current section provides only perfunctory information on this subject, with further details given in the results discussion section (lines 156-163). This arrangement of information is incorrect and needs correction.

Section 5 of the manuscript should begin with a brief presentation of the most relevant results of the study, rather than a reiteration of the purpose of the study and a characterisation of the inclusion and exclusion criteria.  

In addition, the Discussion section repeats from the Results section the figures showing the results obtained. This is completely pointless and needs to be corrected. In the Discussion section of the results obtained, no figures should be included. Furthermore, the authors compare their obtained figures with the results obtained by other authors. It is difficult to say what the results presented in this way testify to. To a large extent, it is rather about the 'accessibility' of the people surveyed, certainly not about the age at which the disease is most prevalent or the gender involved. These data only serve to characterise the study group, possibly as a comparison of one's own results to those obtained in another study and confirmation that a similar study group has been studied

In lines 172-173, the authors referred to 'Asian findings' without providing a reference to these studies. Please provide a citation for these findings.

The data in lines 191-193 of the Discussion section indicate that ‘In the present study 46.0% of the cases had onset of illness in the age group 18-29 years, followed by 33.0% in the 30-39 age group. Additionally, 40.0% of the cases had a duration of illness between 1-3 years, while 25.0% had an illness duration greater than 6 years’. These data should be included in section 4 Results, as they were not previously mentioned.

The footnotes for Figure 1 require additional information to explain the abbreviations 'GI' as well as 'OPD' (line 251).

In Table 1, the patient's income is divided into three categories, but the current labels do not provide enough context for readers unfamiliar with the economic situation of the country. To improve clarity, it may be better to use labels such as low, medium, and high income.

Additionally, the tables take up a significant amount of space compared to the overall text volume, which should be addressed.

Reviewer 3 Report

Comments and Suggestions for Authors

The study evaluates 100 IBS patients aged <60 years from a single tertiary referral center in India. The psychiatric evaluation of patients is carried out with well-validated scales. The population studied presents two relevant characteristics compared to studies carried out in Western countries: the strong prevalence of males and the large prevalence (49%) of subjects without income, despite 51% of graduates. Certainly, these data represent a peculiarity compared to data obtained on large populations representative of the general population in other countries. The doubt of a selection bias, as mentioned in the discussion, is great and these data need to be discussed (relationship between income and symptoms? Between employment status and symptoms in relation to age (students) and sex (housewives)?). In any case, the title must be substantiated by the fact that it refers to a population selected in a tertiary center in India.

Therefore, I believe that the discussion should be reoriented by introducing a comparison with data that is not predominantly local. Furthermore, it is surprising how derived national references are used even for international data. Singular, for example, are the Rome IV Criteria, used for enrollment, it is not provided an original citation but only from derivative publications.

Round 2

Reviewer 2 Report

Comments and Suggestions for Authors

Dear authors, thank you for responding to the review of your manuscript. The revised version you sent incorporated most of the comments.  

Author Response

1) The number of patients screened given in the abstract (150) does not match the number given in the rest of the paper (158).

The number of patients screened were 158. It has been corrected in abstract.

2) The abstract warrants some language editing.

Language editing has been done.

3) The title should be improved in terms of language and precision (as requested by R1), including consistent capitalization of first letters, e. g. Psychiatric Comorbidities in a Clinical Population of Irritable Bowel Syndrome in a Tertiary Care Center in Northern India

The title of the study has been changed to “Psychiatric Comorbidities and Profile of Patients with Irritable Bowel Syndrome in Northern India”

4) Figure 1 looks distorted and should be improved and placed slightly further down in the layout.

The figure has been improved

Reviewer 3 Report

Comments and Suggestions for Authors

Due to the peculiar setting, as now better clarified by the authors, which leads to the study of a population substantially different from that of studies in other parts of the world, the need arises to add the clarification "in India" to the end of the current title, as already suggested in round 1
